# The Efficient Way to Design Cooling Sections for Heat Treatment of Long Steel Products

**DOI:** 10.3390/ma16113983

**Published:** 2023-05-26

**Authors:** Petr Kotrbacek, Martin Chabicovsky, Ondrej Resl, Jan Kominek, Tomas Luks

**Affiliations:** Faculty of Mechanical Engineering, Brno University of Technology, 616 69 Brno, Czech Republic; petr.kotrbacek@vut.cz (P.K.); ondrej.resl@vut.cz (O.R.); jan.kominek@vut.cz (J.K.); tomas.luks@vut.cz (T.L.)

**Keywords:** heat treatment, quenching, heat transfer, heat transfer coefficient, Leidenfrost temperature, cooling section design, steel

## Abstract

To achieve the required mechanical properties in the heat treatment of steel, it is necessary to have an adequate cooling rate and to achieve the desired final temperature of the product. This should be achieved with one cooling unit for different product sizes. In order to provide the high variability of the cooling system, different types of nozzles are used in modern cooling systems. Designers often use simplified, inaccurate correlations to predict the heat transfer coefficient, resulting in the oversizing of the designed cooling system or failure to provide the required cooling regime. This typically results in longer commissioning times and higher manufacturing costs of the new cooling system. Accurate information about the required cooling regime and the heat transfer coefficient of the designed cooling is critical. This paper presents a design approach based on laboratory measurements. Firstly, the way to find or validate the required cooling regime is presented. The paper then focuses on nozzle selection and presents laboratory measurements that provide accurate heat transfer coefficients as a function of position and surface temperature for different cooling configurations. Numerical simulations using the measured heat transfer coefficients allow the optimum design to be found for different product sizes.

## 1. Introduction

The properties of the final product depend on the composition of the steel and the heat treatment process. The microstructure determines the overall mechanical behaviour of the steel and heat treatment provides an efficient way of manipulating the properties of the steel by controlling the cooling rate. When heat-treating steels, it is necessary to achieve the required cooling rate and final product temperature to obtain the required mechanical properties. The cooling strategy is determined by the size and shape of the product. Especially in the steel industry, large products with different sizes (shape and thickness) are heat-treated with the same cooling unit, so the proper regulation of the cooling rate by changing the coolant pressure and flow rate is necessary to achieve good material properties for different product sizes. For these reasons, spray cooling is one of the most common cooling methods used for heat treatment in the steel industry. The design of spray cooling sections is complicated by the existence of different types of nozzles, their proper positioning, sizing, and other parameters affecting the spray cooling of hot surfaces (presence of boiling) [1].

The parameters affecting the spray cooling of hot steel surfaces can be divided into two groups [2,3]. The first group relates to spray parameters such as water mass flux (amount of water sprayed onto the surface), water temperature, water additives and droplet size [3]. The second group relates to the cooled surface: temperature, velocity, roughness, and the presence of oxide scale [3]. The key factors are the water mass flux and the cooled surface temperature [4,5]. The influence of each factor on the heat transfer coefficient (HTC) and/or Leidenfrost temperature is briefly described below to illustrate the complexity of the spray cooling problem in the presence of boiling.

The influence of surface temperature is related to the different boiling regimes (single-phase convection, nucleate boiling, transition boiling and film boiling (Figure 1)) that occur as the hot surfaces cool [6,7]. There are two important points (temperatures) on the boiling curve (Figure 1): the Leidenfrost point (boundary between transition boiling and film boiling) and the critical heat flux point (boundary between transition boiling and nucleate boiling). The heat flux on the boiling curve is at the minimum at the Leidenfrost point and the maximum at the critical heat flux point [6]. The heat transfer coefficient is significantly lower in the film boiling regime than in the nucleate boiling regime. The Leidenfrost temperature is sensitive to the surface condition (scale, roughness), as will be shown later.

The amount of water sprayed onto the cooled surface is commonly expressed as the water mass flux ($\dot{m}$ [kg m^−2^ s^−1^]), which is commonly referred to as the impingement density [4]. Increasing the water mass flux increases the heat transfer coefficient for all surface temperatures [4,8,9]. There are different types of nozzles with different spray footprints (Figure 2). The water mass flux is influenced by the size of the nozzle (which determines the water flow rate) and the spray area (which is influenced by the distance between the nozzle and the cooled surface and by the shape of the jet). The use of nozzles allows a wide range of cooling rates to be achieved by changing the process water pressure. Increasing the water pressure increases the amount of water sprayed, which in turn increases the cooling rate. The Leidenfrost temperature and the temperature at which the critical heat flux occurs increase as the water mass flow increases [10,11].

It is common in the steel industry that the temperature of the cooling water varies during seasons. The water temperature influences the heat transfer coefficient and has a significant effect on the Leidenfrost temperature [13,14]. The paper [15] shows that an increase in cooling water temperature of about 30 degrees Celsius (typical in the industry due to seasonal effects) causes a 10% decrease in the heat transfer coefficient in the film boiling regime and a 17% decrease in the Leidenfrost temperature.

The Leidenfrost temperature (duration of the film boiling regime) can be easily influenced by adding small amounts of additives to the water [16]. Some additives prolong the film boiling regime (solid particles, nano particles [17], gases, oils, and fats) and others shorten it (salts (mostly NaCl), nanofluids [18] and water-soluble acids and alkalis) [16]. Nanofluids and surfactants can increase the heat transfer coefficient during nucleate boiling [19].

The type and size of the nozzle and the water pressure influence the droplet size and velocity. In general, droplet diameter and droplet velocity influence cooling [20,21]. The higher the droplet velocity, the higher the heat transfer coefficient [9].

Surface roughness and the presence of scales (oxides) on the surface influence spray cooling. Higher surface roughness results in a higher heat transfer coefficient [22]. Oxide scales mainly affect the Leidenfrost temperature [23,24] and can increase cooling [25].

Differences in the spray cooling of static and moving plates (typical industrial situation) were reported in [26,27].

The surface temperature-dependent heat transfer coefficient is a necessary input for numerical simulations of the cooling process. At present, some empirical correlations exist for predicting the heat transfer coefficient or Leidenfrost temperature [2,4,9,10,11,28,29], but they are mainly a function of the water mass flux. Due to the many parameters that influence the spray cooling of a hot surface, it is impossible to accurately estimate the heat transfer coefficient by some analytical or empirical equation that takes all these parameters into account. Laboratory measurement provides a fast and accurate method of obtaining the heat transfer coefficient for the spray cooling of hot surfaces.

The process of designing cooling sections for heat treatment is an iterative research process involving several important steps. First, the cooling regime (cooling rate and cooling time) that will provide the desired mechanical properties must be found. Then, the appropriate nozzle types and positions are determined. The cooling section is designed, and the heat transfer coefficient of the designed cooling is measured under different conditions. The next step is to perform numerical simulations showing the optimum cooling settings (water pressure, cooling length and product speeds) for different product sizes (thicknesses). The final step is to validate the designed cooling in a full-scale laboratory or pilot plant test.

## 2. Finding the Optimal Cooling Regime

The first step is to determine the appropriate cooling regime that will provide the required mechanical properties. This can be done by the metallurgist using a Continuous Cooling Transformation (CCT) diagram and numerical simulation of cooling, and/or by quenching tests using small samples and a nozzle [1,30]. These tests are carried out on steel specimens with embedded sheathed thermocouples (typically type K with a diameter of 1.5 mm or less). The schematic diagram of the laboratory test bench, photographs of the tests, and a schematic diagram of the test specimen are shown in Figure 3. 

The thermocouples are positioned at different depths to monitor the temperature across the thickness of the specimen. Holes parallel to the cooled (top) surface (Figure 3C) are drilled from the front of the specimen. Each hole is at a different depth from the cooled (top) surface. Thermocouples are glued into these holes using a special high-temperature ceramic adhesive. This allows good contact (low thermal resistance) between the thermocouple tip and the steel specimen. The measuring positions are in the centre of the sample.

The test procedure is as follows. The sample with embedded thermocouples is heated to a specified temperature and held at that temperature for a specified time. The required water pressure is then set, and the sample is removed from the furnace and placed under the nozzle. The deflector is opened for a specified time and the sample is cooled. After the test, the sample is cut, and the hardness and microstructure are measured. These tests are repeated with different settings (nozzle type, water pressure, cooling time) until the required mechanical properties are found. Once the best solution is found, the cooling section can be designed and tested. Examples of measured data (temperature record, hardness and microstructure obtained) for two different cooling regimes are shown in Figure 4, Figure 5 and Figure 6. Regime 1 had a shorter cooling time and a higher final temperature than Regime 2. The longer cooling time in Regime 2 resulted in a higher hardness and a higher proportion of bainitic structure. The slightly longer cooling in Regime 2 improved the mechanical properties throughout the sample thickness, whereas Regime 1 only improved the first 10 mm. 

## 3. Cooling Section Design: Nozzles Selection and Positioning

It is necessary to know the heat transfer coefficient of different nozzle configurations (type, size, and position) in order to design long cooling sections formed by hundreds of nozzles. Nozzle positions are limited by space constraints, which are influenced by conveyor design, presence of piping, and other technology. The first step (finding the cooling regime) and existing empirical correlations provide rough information on the required water mass flux. This can be used as initial information for nozzle selection. If very intensive cooling is required, it is better to use large flat jet nozzles. In the case of soft cooling, the small full cone nozzles can provide an efficient cooling method. The positions of the nozzles should be chosen according to the space limitations and taking into account the estimated water mass flux. Examples of design are shown in Figure 7 and Figure 8. These figures have been generated using internally developed software (Sim Spray 5.3). This software allows the jets, footprints, and water distribution of different nozzle configurations to be shown. This software includes many types and sizes of nozzles. The water distribution of each nozzle type in this software is taken from laboratory measurements or approximated from catalogue values.

The preliminary designed cooling sections/zones are manufactured in a small, economical version and then tested in the laboratory. The heat transfer coefficients for different nozzle configurations are obtained from these tests. These are used in numerical simulations of the cooling process to validate the required cooling intensity for different products. The preliminary designed cooling section can be modified (optimised) until sufficient cooling homogeneity and the required cooling rate (heat transfer coefficient) are achieved.

## 4. Description of Laboratory Measurement of Heat Transfer Coefficient

Laboratory measurements of the heat transfer coefficient are performed with a moving sample. This allows the heat transfer coefficient to be mapped along the length of the cooling. The schematic is shown in Figure 9, and photos of tests with different products (plate, tube, and rail) are shown in Figure 10.

The test procedure starts with the production of a test sample of the original shape (plate, tube, rail, etc.) and the embedding of thermocouples in the test sample. In the case of a shaped product (tube, rail), the thermocouples are embedded in a similar way (smaller diameter and closer to the surface) as described in Section 2 of this paper, and in the case of a plate, holes are drilled perpendicular to the cooled surface and thermocouples with a diameter of 0.5 mm are inserted into these holes and welded just below the cooled surface. The embedded sensors are calibrated to provide reliable and repeatable heat transfer coefficient values. The sample is connected to the test bench and heated to the desired temperature (typically 900 °C). Once the cooling conditions are set, the data logger begins recording temperatures and sample position. The sample is moved reversibly through a cooling zone until it has cooled to room temperature. The maximum speed of movement on the test bench is 10 m s^−1^. The recorded data are then transferred to a computer for analysis. 

Cooling uniformity is critical to design. During testing, cooling uniformity can be measured using thermocouples in different positions or by using line scanners or infrared cameras. A typical output from a line scanner measurement is shown in Figure 11. The tests on flat products are carried out for top and bottom cooling, and the optimum setting of the bottom cooling should be found to provide the same top and bottom cooling.

After the measurement, the inverse heat conduction problem is solved to calculate the time-dependent boundary conditions. The inverse heat conduction problem [32] is a task where we do not know the boundary conditions (heat flux, heat transfer coefficient), but we do know the time-dependent temperatures measured at internal points of the body (temperature records from individual thermocouples). The basic inverse problem for the one-dimensional case can be written in the form [32]:(1)$$\frac{\partial }{\partial x}\left(\lambda \frac{\partial T}{\partial x}\right)=\rho c_{p}\frac{\partial T}{\partial t},$$
$$T\left(x,0\right)=T_{0}\left(x\right),\frac{\partial T}{\partial x}=0,forx=L,T\left(x_{l},t_{i}\right)=T_{i}^{*}.$$
where Equation (1) is a partial differential equation of the heat conduction in one dimension (*T*—temperature, *x*—position, *t*—time, λ—thermal conductivity of the steel, *ρ*—density of the steel, *c_p_*—specific heat of the steel, *L*—thickness of the steel specimen, *T*_0_(*x*)—initial (*t* = 0) temperature field in the steel specimen, *x_l_*—distance between measuring point and cooled surface (*x* = 0), $T_{i}^{*}$—temperatures measured by thermocouple in time steps *t_i_*, All in SI units). The scheme of the model is shown in Figure 12.

The objective is to determine the heat flux (heat transfer coefficient and temperature) at the surface at time steps *t_i_*:(2)$$\dot{q}\left(t_{i}\right)={\left.-\lambda \frac{\partial T}{\partial x}\right|}_{x=0}$$

To solve the inverse problem in this work, the sequential identification method described in [1,33,34] was used. In this method, the heat transfer coefficient at the surface is calculated step by step based on the measured temperatures from the thermocouples. The method uses a sequential estimation of time-varying boundary conditions and forward time steps to stabilise the inverse problem. The heat transfer coefficient in time *m* is obtained by minimising Equation (3) (the same equation as in the classical Beck approach [32]):(3)$$SSE=\sum_{i=m+1}^{m+r}{\left(T_{i}^{*}-T_{i}\right)}^{2}$$

This method allows the heat transfer coefficient to be calculated even when the homogeneity of the surface temperature is disturbed by built-in thermocouples.

An example of the calculated time-dependent surface temperature and heat transfer coefficient is shown in Figure 13. The HTC as a function of the surface temperature and the position in the cooling section is obtained by interpolation (Figure 14). HTC as a function of surface temperature can be obtained by averaging along the position (Figure 15—left). This boundary condition can be easily applied in any commercial FEM software. The dependence of HTC on position (Figure 15—right), obtained by averaging along the surface temperature axis, can be useful for optimising nozzle positions. The dependence of the HTC on position is significantly influenced by the position of the nozzles (the main factor influencing the cooling is the water mass flux). The peaks in Figure 15 (right) are at the positions of the nozzles (maximum water mass flux) and between the nozzles, and where no water is sprayed, the heat transfer coefficient reaches its minimum. The design must take into account that a large distance between the nozzles can cause a significant decrease in the heat transfer coefficient between the nozzle positions.

The heat transfer coefficient of the selected configuration depends on the water pressure used. An example of the effect of pressure is shown in Figure 16. The difference between two and six bars is significant for higher surface temperatures (Figure 16). The heat transfer coefficient for six bars is almost double that for two bars at 600 °C. The difference is less significant at lower surface temperatures. The cooling process of the installed cooling section can be controlled by changing the coolant pressure, especially at high surface temperatures.

## 5. Simulation of the Real Cooling Process

The cooling process can be numerically simulated using the boundary conditions known from the previous step. The simple 1D finite element method (FEM) simulation can be used for thin flat products. Otherwise (shaped profiles, rails, etc.), the 2D FEM model should be used to simulate the cooling process. The ideal boundary condition is the heat transfer coefficient as a function of surface temperature and position in the cooling section. The laboratory boundary conditions obtained with the short cooling section can be combined to form a boundary condition for the long cooling section (Figure 17).

The results of numerical simulations with different settings are the cooling curves for different conditions (product size, product speed, cooling water pressure, etc.). These cooling curves are then compared, and the optimum cooling settings are selected for different product sizes.

The final step is to validate the mechanical properties of the heat-treated material. The best way to do this is to carry out expensive pilot tests in the plant/rolling mill. Laboratory tests with rotating samples can be an inexpensive way to simulate continuous cooling under plant conditions with a high product speed and long cooling section. Straight motion is converted into rotation. A schematic diagram and photo of a laboratory test bench is shown in Figure 18. It consists of nozzles, a deflector, and a rotating arm with a hot test sample. This test bench allows for switching between two types of nozzles to simulate an initial intensive cooling followed by a soft cooling. An example of the temperatures measured on this test bench for the one-sided cooling of a 20 mm thick plate (simulation of two-sided cooling 40 mm) is shown in Figure 19. The result of the numerical simulation for this cooling regime (simulation was carried out before this measurement) is added to this graph and it shows good agreement with the measurement. The results of measured hardness, tensile and Charpy pendulum tests of this heat-treated material (real cooling conditions) are shown in Figure 20 and Figure 21.

## 6. Conclusions

The design technique presented combines numerical simulations, laboratory measurements and, in the final stage, testing under production conditions. Such a technique minimises the number of expensive pilot trials and avoids potential design errors. Optimum design is achieved by accurately measuring the heat transfer coefficient of the designed cooling system. This allows for the accurate numerical modelling of the cooling process. This method can be used to design new cooling sections as well as to optimise existing cooling. An optimised design based on laboratory measurements of the heat transfer coefficients allows the required mechanical properties to be achieved without the need for complex on-site testing, and the new cooling section can be commissioned more quickly and without the need for additional cooling modifications. The optimised design saves water, pumping energy, and manufacturing costs by optimising the length of the cooling section and the number of nozzles.

## Figures and Tables

**Figure 1 materials-16-03983-f001:**
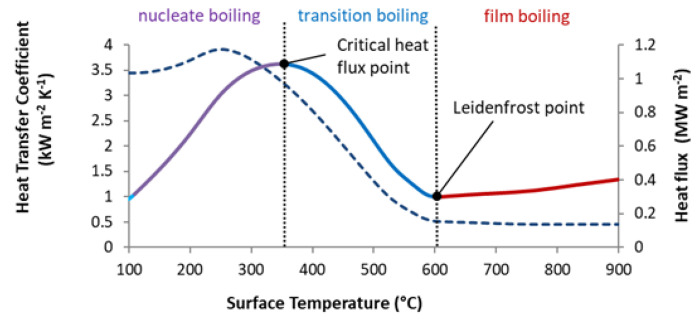
Dependences of the heat transfer coefficient (dashed line) and heat flux (solid line) on the surface temperature [3].

**Figure 2 materials-16-03983-f002:**
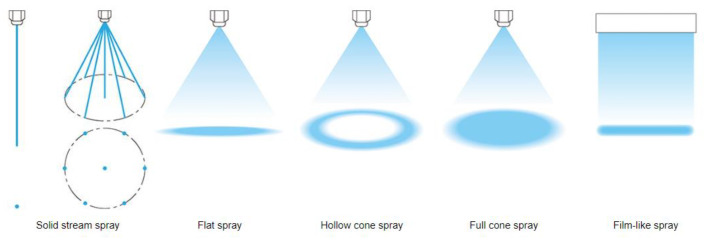
Different jet shapes and their footprints [12].

**Figure 3 materials-16-03983-f003:**
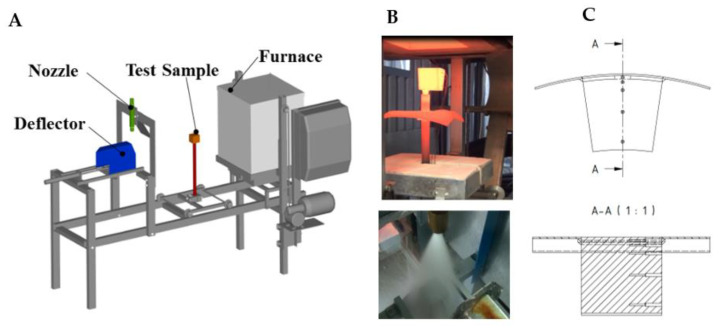
Laboratory heat treatment tests—(**A**): stationary spray quenching test bench [30], (**B**): photos of tests, (**C**): Schematic of test specimen with holes for thermocouples.

**Figure 4 materials-16-03983-f004:**
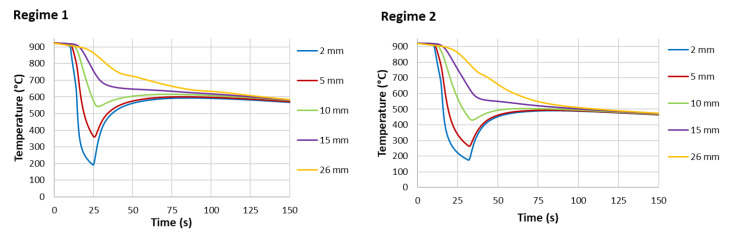
Example of measurement results for two different cooling regimes and a sample thickness of 30 mm, steel grade S355: cooling curves at different positions from the cooled surface.

**Figure 5 materials-16-03983-f005:**
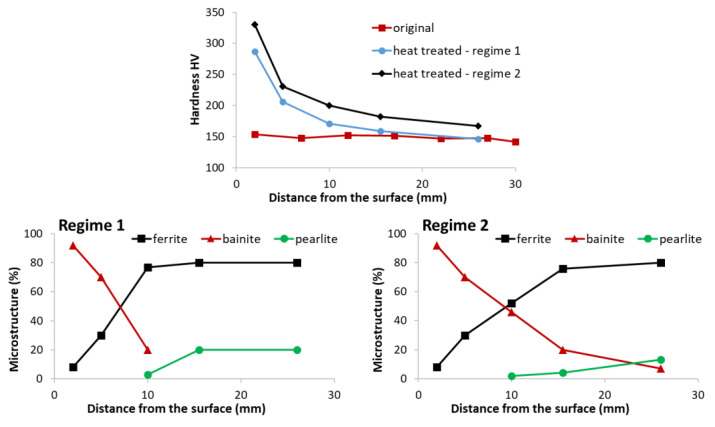
Example of measurement results for two different cooling regimes and a sample thickness of 30 mm, steel grade S355: measured hardness—Vickers HV30 (**top**) and microstructure composition (**down**).

**Figure 6 materials-16-03983-f006:**
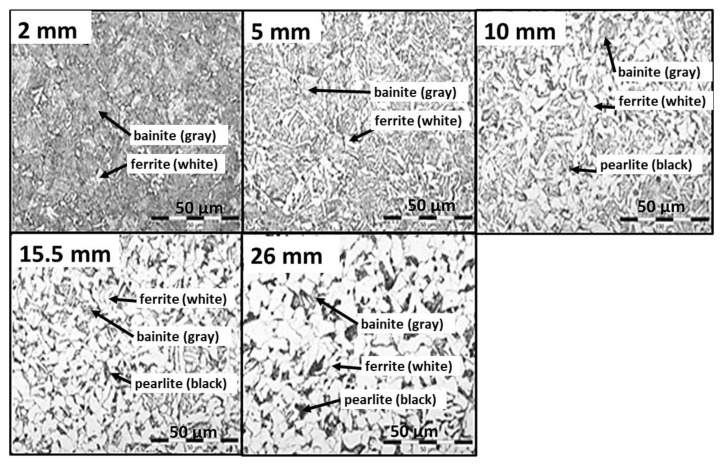
Example of measurement results for cooling Regime 2 and a sample thickness of 30 mm, steel grade S355: microstructure analysis.

**Figure 7 materials-16-03983-f007:**
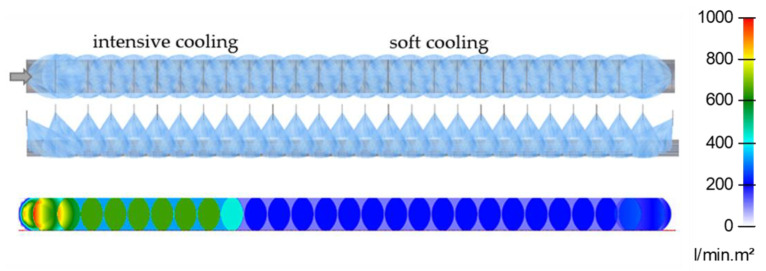
Example of preliminary design for steel flat strip: **top**—nozzles positions (top and side view), **bottom**—water distribution.

**Figure 8 materials-16-03983-f008:**
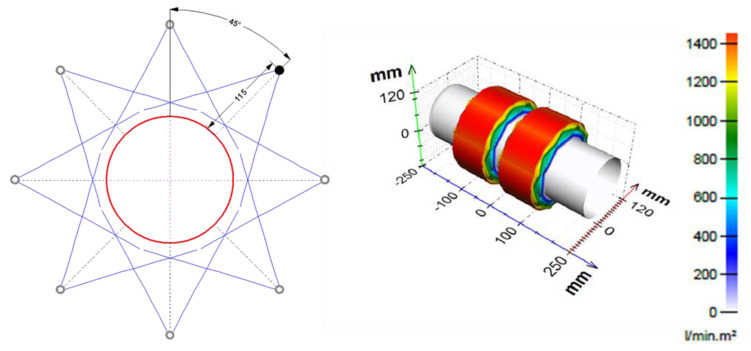
Example of preliminary design for steel tubes: **left**—nozzles positions, **right**—water distribution.

**Figure 9 materials-16-03983-f009:**
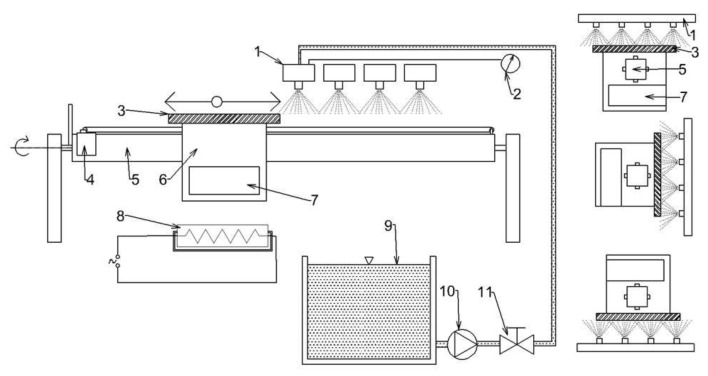
Schematic of the laboratory test bench for measuring the heat transfer coefficient: 1—headers with nozzles, 2—pressure gauge, 3—test plate, 4—motor moving trolley, 5—girder carrying trolley, 6—movable trolley, 7—data logger, 8—heater, 9—water tank, 10—pump, 11—control valve [31].

**Figure 10 materials-16-03983-f010:**
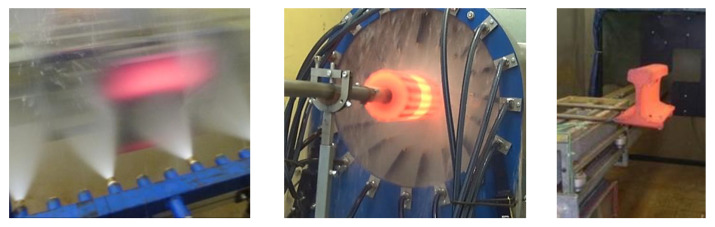
Photos of laboratory stand with tested cooling for plate, tube, and rail.

**Figure 11 materials-16-03983-f011:**
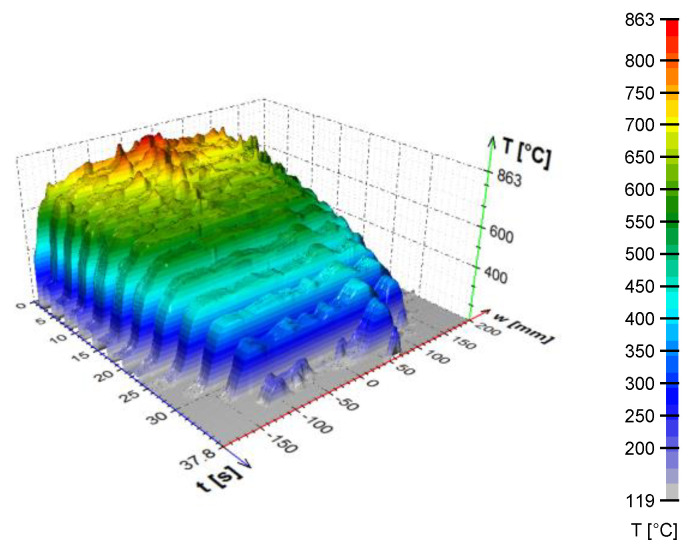
Example of typical results obtained with the line scanner.

**Figure 12 materials-16-03983-f012:**
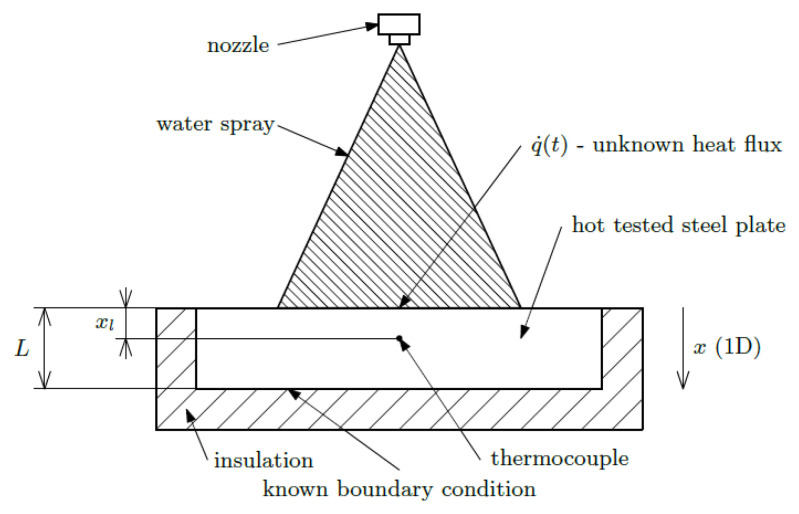
Scheme of one-dimensional inverse heat conduction problem.

**Figure 13 materials-16-03983-f013:**
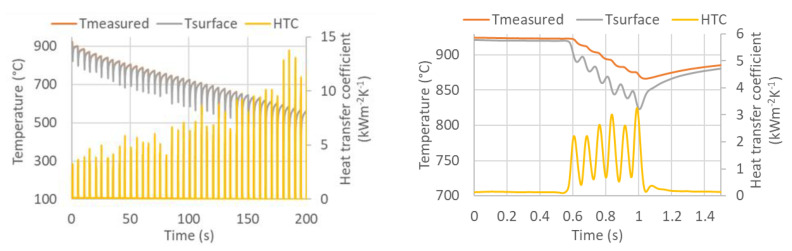
Measured temperature, computed surface temperature, and computed heat transfer coefficient (**left**: all experiment, **right**: detail of the first pass through the cooling section).

**Figure 14 materials-16-03983-f014:**
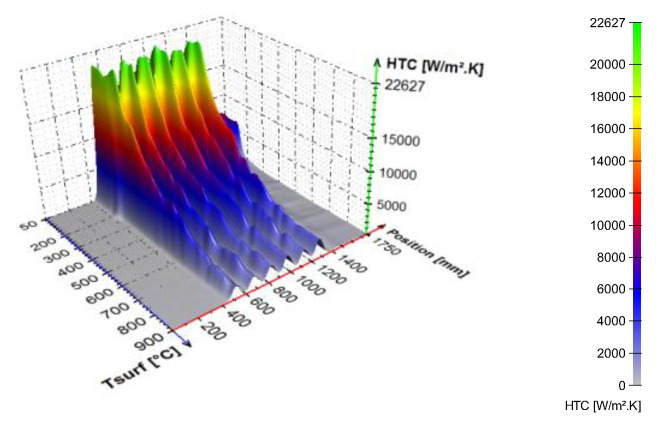
HTC as a function of position and surface temperature.

**Figure 15 materials-16-03983-f015:**
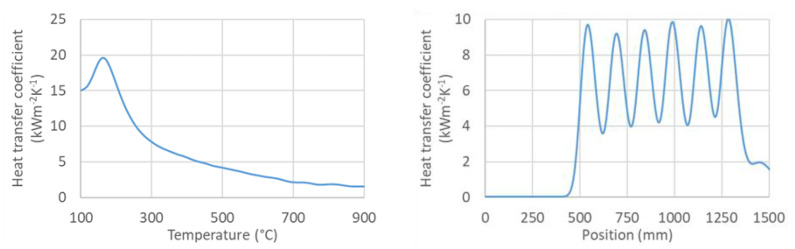
(**Left**): HTC as a function of surface temperature, (**right**): HTC as a function of position (50–900 °C).

**Figure 16 materials-16-03983-f016:**
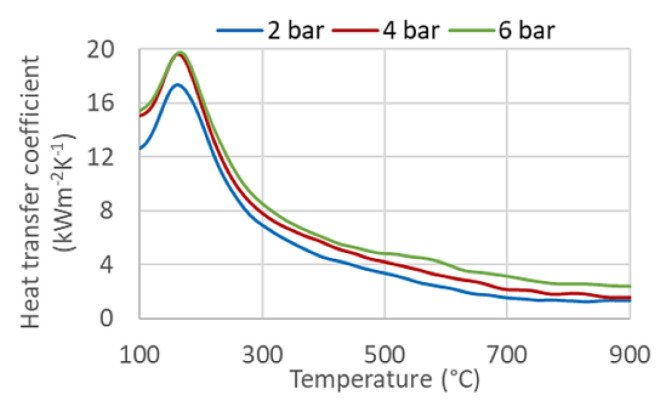
Influence of the water pressure on the heat transfer coefficient.

**Figure 17 materials-16-03983-f017:**
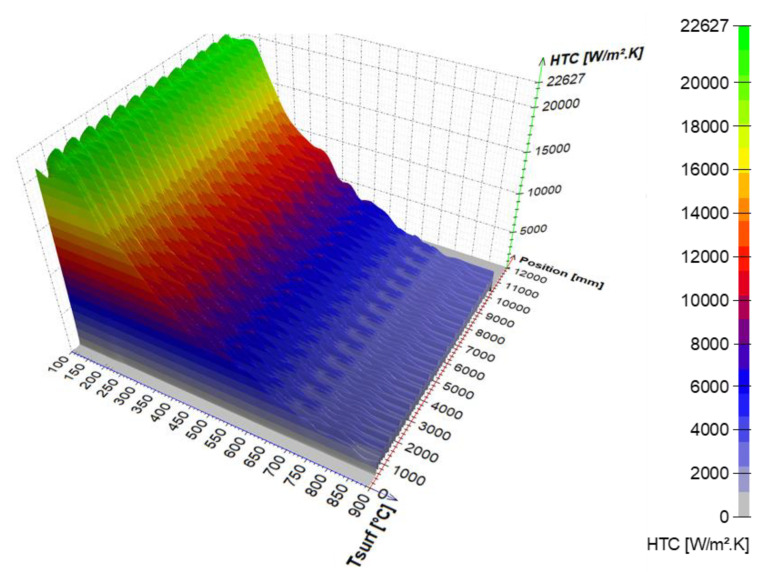
Combined boundary condition (12 m).

**Figure 18 materials-16-03983-f018:**
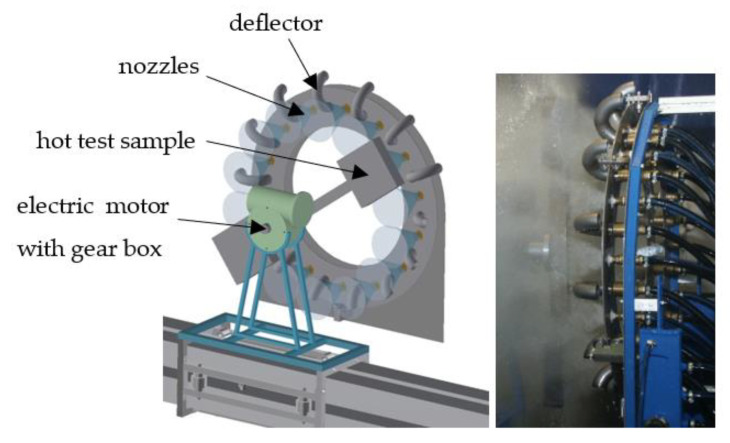
Laboratory test bench with rotating sample.

**Figure 19 materials-16-03983-f019:**
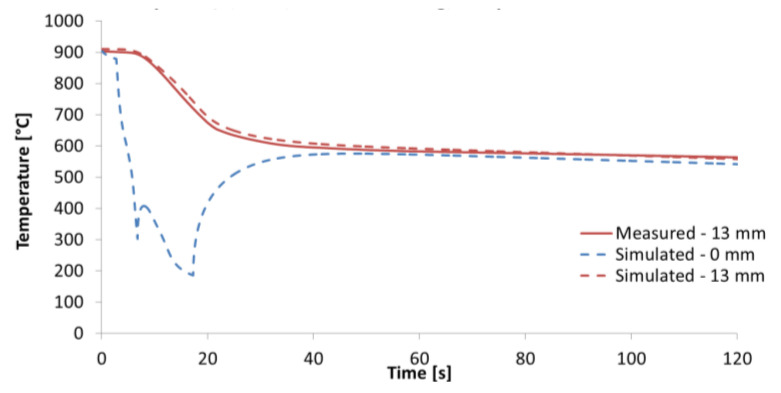
Example of measured temperatures (test bench with rotating sample) for one-sided cooling of a 20 mm thick plate of S355 steel (simulation of two-sided cooling of a 40 mm thick plate). Cooling regime: Intensive cooling—8 bar, length 16 m (4 s) and then soft cooling—4 bar, length 42 m (10.5 s), product speed 4 m s^−1^. Total length of cooling section—58 m (14.5 s).

**Figure 20 materials-16-03983-f020:**
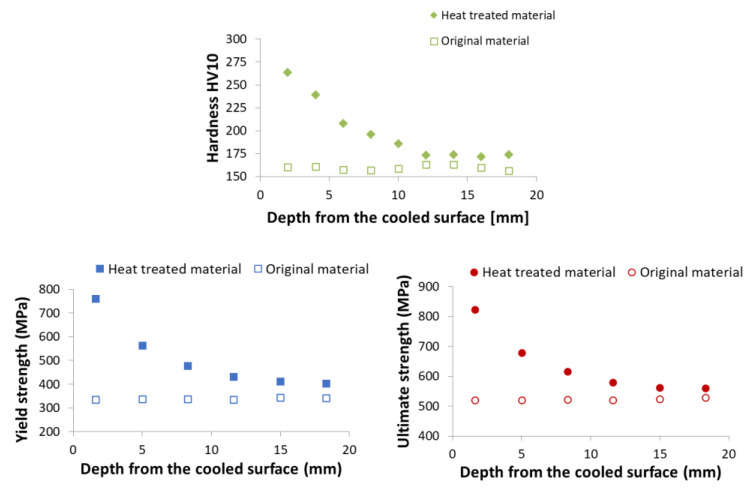
Measured Vickers hardness for material S355: original material and heat-treated (regime shown in Figure 19).

**Figure 21 materials-16-03983-f021:**
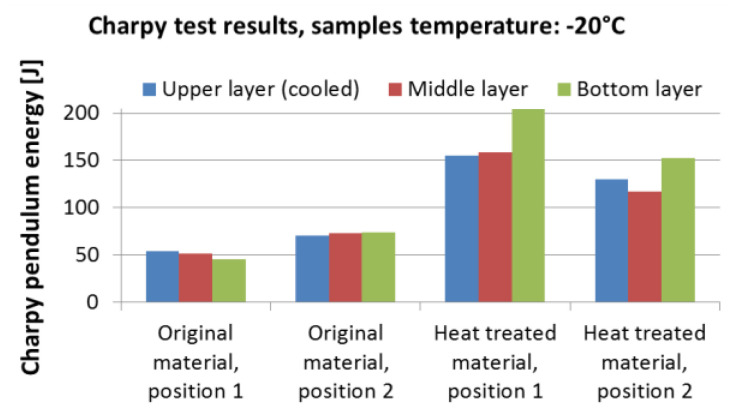
Results of Charpy pendulum tests for material S355: original material and heat-treated (regime shown in Figure 19).

## Data Availability

The data presented in this study are available on request from the corresponding author.

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
