# Peer review of "The Efficient Way to Design Cooling Sections for Heat Treatment of Long Steel Products"

_materials, 2023, doi:10.3390/ma16113983_

Round 1
Reviewer 1 Report
(1) In Fig.6, the different phase should be labeled.
(2) How to embed thermocouples at different depths? Please describe this in details.
(3) How to acquire Figure 7 and Figure 8?
(4) The solution of inverse heat conduction problem should be described deeply.
(5) What is the inner mechanisms of the dependence of HTC on position? What is the fundamental influencing factor?
Reviewer 2 Report
There are two remarks I want to point out in the draft:
1. In some places in the manuscript, the temperature is expressed in Kelvin (line 78) and in some places in degrees Celsius (Fig. 4). This should be standardized and only one scale (Celsius) should be used.
2. There are too many self-citations in the manuscript. Out of 25 references, 10 are self-citations (40%). Considering that M. Raudensky [9, 10, 16, 20] works in the same research team, there are 4 more citations of him (which gives 14 self-citations – 56%). Also other researchers in the world deal with this subject and it is worth quoting their works.
